# Efficacy of Overground Robotic Gait Training on Balance in Stroke Survivors: A Systematic Review and Meta-Analysis

**DOI:** 10.3390/brainsci12060713

**Published:** 2022-05-31

**Authors:** Matteo Lorusso, Marco Tramontano, Matteo Casciello, Andrea Pece, Nicola Smania, Giovanni Morone, Federica Tamburella

**Affiliations:** 1Santa Lucia Foundation, Via Ardeatina 306, 00179 Rome, Italy; m.lorusso@hsantalucia.it (M.L.); matteo1990casciello@gmail.com (M.C.); f.tamburella@hsantalucia.it (F.T.); 2Department of Movement, Human and Health Sciences, University of Rome “Foro Italico”, 00185 Rome, Italy; 3Ospedale Israelitico di Roma, Via Fulda 14, 00148 Rome, Italy; andrea.pece1973@gmail.com; 4Neurorehabilitation Unit, University Hospital of Verona, 37124 Verona, Italy; nicola.smania@univr.it; 5Department of Life, Health and Environmental Sciences, University of L’Aquila, 67100 L’Aquila, Italy; giovanni.morone@univaq.it

**Keywords:** stroke, balance function, overground exoskeleton, overground robot-assisted gait training

## Abstract

Strokes often lead to a deficit in motor control that contributes to a reduced balance function. Impairments in the balance function severely limit the activities of daily living (ADL) in stroke survivors. The present systematic review and meta-analysis primarily aims to explore the efficacy of overground robot-assisted gait training (o-RAGT) on balance recovery in individuals with stroke. In addition, the efficacy on ADL is also investigated. This systematic review identified nine articles investigating the effects of o-RAGT on balance, four of which also assessed ADL. The results of the meta-analysis suggest that o-RAGT does not increase balance and ADL outcomes more than conventional therapy in individuals after stroke. The data should not be overestimated due to the low number of studies included in the meta-analysis and the wide confidence intervals. Subgroup analyses to investigate the influence of participant’s characteristics and training dosage were not performed due to lack of data availability. Further well-designed randomized controlled trials are needed to investigate the efficacy of o-RAGT on balance in individuals with stroke.

## 1. Introduction

Stroke is the third leading cause of death after cardiovascular diseases and cancer. Moreover, stroke is the world’s leading cause of disability [1], with a high prevalence of ischemic etiology (85%) which is the result of a transient or permanent reduction in blood flow in the territory of a cerebral artery [2].

Patients with stroke often experience deficits in motor control that contribute to a reduced balance function [3]. The balance function is the ability to maintain the center of gravity within the base of support with minimal postural sway [4] and this can be achieved by a complex multifactorial system, consisting of sensory, motor, visual and cognitive components, interacting with the environment [5]. Balance dysfunction in individuals with stroke can have a negative impact on mobility and increase the risk of falls [6], thus reducing autonomy and independence in the activities of daily living (ADL). Balance recovery is considered an important factor in achieving independent walking and is also a significant predictor for gait function [7]. Training and practice on balance control strategies can improve balance and gait, which are the main goals of neurorehabilitation programs to restore effective and safe mobility [8]. However, when considering the determinants of independent mobility, gait receives more attention than balance function due to its association with ADL, but walking is only possible with the ability to maintain stability [9].

In recent years, powered robotic devices have been introduced in stroke treatment to maximize the recovery of individuals with stroke [10]. Robot-assisted gait training (RAGT) devices can provide repetitive task training, leading to functional recovery and improving motor control in patients with stroke [8]. Commercially available devices are commonly divided into stationary systems (s-RAGT) and overground systems (o-RAGT) [11]. The former is implemented by using a fixed structure combined with a moving ground platform. The s-RAGT can be distinguished into treadmill-based gait trainers with exoskeleton (t-RAGT) and end-effector gait trainers [11]. The t-RAGT is a device (driven by a motor) with an endless belt on which the patient walks and in which the movement of the leg is produced by the exoskeleton worn by the patient. The end-effector gait trainer is a device with two independently moving footplates, onto which the patient’s feet are fixed. The movements of the plates induce the stance and swing phases of the patient’s gait. Both devices are used in association with a bodyweight support (BWS) system [11], although the treadmill may be used without a BWS. t-RAGT with BWS is the most widely used approach in gait neurorehabilitation [12].

The o-RAGT are robotic devices that allow patients to practice gait on a hard surface. Steps are activated by the therapist’s control or by the patient through a trigger based on weight or trunk shifting [11,13,14]. Moreover, these overground walking devices allow the execution of postural and balance exercises [11]. Although the primary purpose of RAGT is to train walking, these types of training imply also a continuous involvement of equilibrium control that may indirectly improve patient’s balance ability [15,16]. Thus, it is surprising how few studies have reported the efficacy of RAGT on balance function. In particular, one study reported balance outcomes when patients were trained with s-RAGT [16], another two studies with BWS [15,17] and another one with mixed overground devices comprising exoskeleton and robotized orthosis also assisted by BWS [18]. No data are available on ADL in these reviews or in the framework of stroke o-RAGT.

Despite the widespread use of overground exoskeletons in the field of stroke rehabilitation, there have been no reviews specifically focusing only on o-RAGT efficacy on balance. Therefore, the primary aim of this systematic review and meta-analysis is to explore the existing literature on o-RAGT efficacy on balance function in people with stroke. Moreover, the secondary aim is to analyze the efficacy of o-RAGT on ADL in the same population. Furthermore, the influence of training dosage (frequency, intensity and duration), the epidemiological and clinical features of the participants on balance and ADL after o-RAGT are investigated.

## 2. Materials and Methods

This systematic review and meta-analysis was performed according to the PRISMA (Preferred Reporting Items for Systematic Reviews and Meta-Analyses) statement [19] and the protocol registered in the PROSPERO database in December 2021 (CRD42022295736).

### 2.1. Eligibility Criteria

Studies in which the population of interest was adults (age > 18) with a history of ischemic or hemorrhagic stroke were included. No restrictions were applied on sex or time since stroke. The interventions considered were rehabilitation training with the use of an overground exoskeleton, alone or in association with conventional therapy (CT), with no restrictions on the number of sessions provided. Comparison interventions included CT or other technological devices. Both clinical and objective instrumental assessments were considered as outcome measure of balance. The search was limited to full-text studies published in English and on human participants. Controlled and non-controlled clinical trials (i.e., Randomized Clinical Trials (RCTs) and non-RCTs), retrospective studies, case series, case reports and observational studies were considered eligible. Restriction on publication date was not applied.

Exclusion criteria were studies involving individuals with neurological diseases other than stroke, or which involved the hybrid application of the overground exoskeleton (e.g., functional electrical stimulation, transcranial magnetic stimulation, transcranial direct current stimulation) or providing exoskeleton training in association with a treadmill or with the use of BWS. No peer-reviewed papers, conference proceedings, congresses’ abstracts, editorials, letters or reviews were excluded.

### 2.2. Data Sources and Searches

The research was conducted from inception until 30 November 2021, in the following databases: MEDLINE (Medical Literature Analysis and Retrieval System Online), PEDro (Physiotherapy Evidence Database), the Cochrane Central Register of Controlled Trials, Web of Science and Scopus. The following search strategy was used: (stroke OR “cerebrovascular accident” OR “cerebral stroke”) AND balance AND (robot* OR exoskelet* OR “exoskeleton device”). The same search strategy was conducted in each database. The only exception was that, in the PEDro database, keyword terms were combined to obtain records. In addition, manual searches were performed in the reference lists of the retrieved articles and previous published reviews or meta-analyses.

### 2.3. Study Selection and Data Collection Process

After the removal of duplicate records, two independent assessors (M.L. and M.C.) reviewed the titles and abstracts considering the established eligibility criteria, and then reviewed the full text of the eligible articles. In the case of discrepancy, a third reviewer (F.T.) was consulted to resolve it. Data extraction of the following relevant features of the included studies was performed, using a predefined data extraction form: authors, title, year and country of publication, study design, individuals features (number of participants, sex, mean age, time since stroke, etiology, hemiparesis side and ability to walk independently or not), type of o-RAGT device, interventions data (single session duration, frequency, total number of session, total duration and follow-up), clinical scales and/or instrumental outcome measures, results, drop-out participants and adverse events.

### 2.4. Risk of Bias Assessment

The methodological quality score of all the studies included was calculated according to the recognized Downs and Black (D&B) tool [20], which is organized in different subsections: Reporting, External Validity, Internal Validity (bias) and Internal Validity (confounding). The total score ranges from 0 to 28, in which the higher the score, the higher the methodological quality. In fact, a score below 11 points indicates “poor” quality; 11–19 points reflect “moderate” quality; a score above 19 points is considered “good” quality. All the studies included were assessed according to the D&B tool by two independent reviewers (M.L. and M.C.) to determine the methodological quality score. Score discrepancies were resolved through discussion with a third author (F.T.).

The Cochrane Risk of Bias 2 (RoB 2) tool [21] was also used to assess the risk of bias in controlled trials.

### 2.5. Data Synthesis

Data analysis was performed with the Review Manager version 5.4.1 software (Cochrane, London, UK). The authors of the studies were contacted for further information in the case of missing data, including the mean and standard deviation of the outcome of interest. In addition, individual participant data were requested in order to carry out the subgroup analysis. When the mean and standard deviation were not reported, they were calculated from the median and interquartile data, as indicated by Wan et al. [22]. A meta-analysis was performed to compare the post-intervention changes between the experimental group (o-RAGT with or without CT) and the control group when at least 3 RCTs [15] were available that provided the same treatment to the control group and measured changes using the same clinical scale for both assessments, balance and ADL. The studies were grouped according to the outcome measure and the standardized mean differences (MDs) were calculated, together with their 95% confidence intervals (CIs). An I^2^ value > 40% was considered as the threshold for statistical heterogeneity [23]. Subgroup analyses regarding the influence of training dosage (frequency, intensity, duration and type of device), epidemiological and clinical features of participants on balance after o-RAGT were not reported due to insufficient data availability, despite having been requested.

In addition to the meta-analyses, a descriptive synthesis was performed for the outcomes where statistical pooling was not possible, and the findings are presented in a narrative form with complementing tables.

## 3. Results

### 3.1. Study Selection

From the considered databases and the manual search of the other systematic reviews, a total of 1309 records was identified; 571 of these records were removed because they were duplicates.

Title and abstract screening of the remaining 738 records was completed with the following results: 703 records were excluded and 35 records were considered eligible. After full-text analysis, 26 out of the 35 eligible articles were excluded for the following reasons: treadmill or body weight-supported intervention (n = 21), full text not in English (n = 4) and no balance outcome (n = 1). Consequently, nine articles were included in this systematic review and four studies were included in the quantitative synthesis (meta-analysis). See Figure 1 for the PRISMA flowchart of the study selection process.

All the included studies were published between 2015 and 2021. The studies were conducted in various countries: two in Japan [24,25], two in Russia [26,27] and one in Italy [28], Sweden [29], Poland [30], Canada [31] and Spain [32].

### 3.2. Methodological Quality Assessment

The included studies were mainly RCTs, of which there were six out of nine [27,28,29,30,31]. Moreover, two pilot studies [24,32] and one case-report study [25] were included.

D&B tool average total score across the nine included articles was 17.4 (± 5.6) out of 28. A total of four RCTs [28,29,30,31] were classified with good quality scores. The D&B tool score of a pilot study [32] indicates poor quality, whereas the remaining four studies [24,25,26,27] had moderate scores. D&B tool scores were reported in descending order in Table 1.

A summary of the risk of biases using the RoB 2 tool [22] is reported in detail for each RCT in Figure 2. It was generated with the Review Manager Version 5.4.1 software.

The assessment of the risk of bias showed that a low risk of bias was identified with regard to selection, attrition and reporting bias. On the contrary, as could be expected, since these were rehabilitative interventions, it was not possible in any of the studies to maintain the blindness of patients for the assignment. Finally, the results are conflicting with regard to the concealment of patient allocation. In fact, in three studies, it is not known whether randomization was performed before or after the assessment, so that a high risk of selection bias was revealed, and in two of these studies, the blindness of the assessors was not declared. For this reason, a high risk of selection bias was reported.

### 3.3. Participants

A total of 273 participants was included from all the studies, of whom 133 were males and 74 were females. Data on the gender of 42 participants recruited in one study [27] and 24 participants who dropped out were not available. The average age of the participants was classified according to various age ranges: seven studies [24,25,26,27,29,31,32] focused on participants in the range of 45–64 years, while the remaining two studies [28,30] recruited participants older than 65 years of age. Based on the time elapsed since the stroke, five studies [24,26,27,29,31] focused on individuals in the subacute phase of recovery, whereas three studies [25,28,32] observed the efficacy of o-RAGT in individuals in the chronic phase of recovery. Considering this classification, individuals involved in the included studies were 172 in the subacute phase and 44 in the chronic phase. In addition, the remaining study recruited a mixed population of sub-acute and chronic stroke [30], with no participant details. Focusing on etiology, 217 individuals suffered an ischemic stroke and 29 a hemorrhagic stroke; 124 individuals had a left hemiparesis, while 83 a right one. Bortole et al. [32] did not report data about etiology; instead, data on the hemiparesis side were not reported by Kotov et al. [27]. In addition, two studies [29,31] measured the efficacy of o-RAGT on a total of 64 ambulatory dependent participants and one study [25] was conducted on one independent ambulatory participant. Three studies [26,27,28] were conducted in a mixed population of ambulatory dependent and independent individuals. Lastly, two studies [30,32] did not report participant information about walking abilities. Data about demographic and clinical features are reported in Table 2.

### 3.4. Intervention

The exoskeletons included were EksoGT (n = 3) [28,30,31], HAL (n = 3) [24,25,29], ExoAtlet (n = 2) [26,27] and H2 (n = 1) [32] devices.

Among the studies, seven [24,25,26,28,29,30,31] were conducted by associating o-RAGTs, with CT while in the remaining two studies [29,31], participants received o-RAGT alone.

No adverse events were reported during o-RAGT in the included studies. The only exception was Calabrò et al. [28] who reported a mild skin bleachable erythema in seven individuals.

The intervention data on single session duration, frequency, total number of sessions and total duration differed among the studies and are shown in Table 3. The total training duration varied from 2 to 8 weeks, with a frequency ranging from 1 to 5 times per week. The duration of the single training session varied from 10 to 60 min and the total number of sessions ranged between 8 and 40.

### 3.5. Comparison

Three studies [24,25,32] did not compare exoskeletons trainings with other interventions. In five RCTs, the control group underwent CT [27,28,29,30,31]. For three out of five studies [28,30,31], the dosage of intervention was the same for both groups; for one study [27], it was greater for the experimental group and for another [29], this information was missing. In a single RCT [26], the control group received a cyclo-ergometer intervention (Cy-E).

Regarding follow-up examinations, these assessments were performed by three out of nine studies (n = 1 for EksoGT; n = 2 for HAL): 6 months after stroke onset [31] and after 2 [25] or 6 months after training suspension [29].

### 3.6. Outcome Measure

Of the studies included in this review, two (n = 2 EksoGT [28,30]) aimed primarily at assessing the efficacy of o-RAGT on balance. For the remaining seven studies, balance was assessed as a secondary aim. In addition to balance, all studies evaluated o-RAGT efficacy also on other domains of interest, such as severity of impairments, spasticity, strength, cardiovascular parameters, Quality of Life (QoL), ADL and cognitive impairments. Nonetheless, according to the primary and secondary aims of this systematic review and meta-analysis, only data on balance and ADL were reported (see Table 4).

Among the included studies, the most widely used clinical scale for balance assessment was the Berg Balance Scale (BBS) (n = 1 EksoGT [31]; n = 3 HAL [24,25,31]; n = 2 ExoAtlet [26,27]; n = 1 H2 [32]), whereas the Timed Up and Go (TUG) test was used for balance assessment in three studies, either alone (n = 1 EksoGT [28]) or in combination with BBS (n = 1 HAL [25]; n = 1 H2 [32]. In addition to clinical scales, an instrumental balance assessment was performed in three studies (n = 1 Ekso GT [30]; n = 2 ExoAtlet [26,27] (see Figure 3a): using a static stabilometric system in two studies [26,27] and a baroresistive platform in the other [30].

Three [28,30,31] out of six RCTs evaluated the efficacy of EksoGT device training: one did not reported any significant change [31] and two of them reported an improvement in balance function [28,30]. Both studies defined as primary aim the assessment of the efficacy of o-RAGT on balance function: measuring it in one study with the TUG [28] and in the other one with the baroresistive platform as instrumental assessment [30]. In the former study, 8 weeks of training allowed TUG improvement in the experimental group, whereas no significant change was reported for the control group. In the latter study, the instrumental assessment revealed different results for open-eye (OE) or closed-eye (CE) conditions between groups after 4 weeks of training: center of pressure (COP) X and Y deviation improved for the OE condition for experimental group, whereas COP Y deviation improved for the CE condition after CT. However, a baseline Y deviation difference for the OE condition was reported in the comparison between the two groups. This difference was retained in the comparison performed at the end of training. In addition, at the end of the training, COP path length (L) and mean speed (V) were significantly lower for the experimental group and, for the CE condition, data for Y deviation were significantly higher for the experimental group.

The two RCTs focusing on the ExoAtlet device compared o-RAGT efficacy with a control group undergoing CT [27] or the Cy-E Ortorent MOTO Pedal Trainer [26], both assessing the balance function with a clinical scale and a static stabilometric system. In both studies, significant changes in the BBS and COP data (L, surface area (S) and energy index (Ei)) were reported in both groups, but the comparison between groups showed greater improvements for the experimental ones.

The remaining RCT focusing on the HAL device [29] did not report significant improvements of balance, but revealed a significant positive correlation between the BBS score and the self-perceived mobility of the Stroke Impact Scale (SIS) questionnaire in the experimental group.

In the pilot study [24] and in the case-report study [25] focusing on the HAL device and in the pilot study using the H2 device [32], balance was addressed only using clinical scales, but no significant data were reported. Nevertheless, a positive trend of balance function improvement was observed in these studies.

Regarding the secondary aim, four RCTs out of nine studies addressed o-RAGTs efficacy on ADL: three studies selected the Barthel Index (BI) (n = 1 EksoGT [30]; n = 2 ExoAtlet [26,27]) and in the remaining one the SIS questionnaire was administered (n = 1 HAL [29]) (see Figure 3b). Significant improvements were reported in two RCTs (n = 1 EksoGT [30] and n = 1 ExoAtlet [26]) assessing changes in functional status per the BI. These improvements were reported either for the experimental or control groups. Nevertheless, the BI improvement was higher for the o-RAGT groups than the CT group [30] and Cy-E training group [26].

### 3.7. Meta-Analysis

According to the inclusion criteria for meta-analysis, it was conducted on three RCTs focusing on BBS for the balance assessment (see Figure 4a), and on three RCTs focusing on BI for ADL evaluation (see Figure 4b). Figure 4 shows the results of the meta-analysis carried out by comparing BBS and BI scales. The mean, standard deviation (SD), total number of participants and data for continuous variables were reported as the mean difference, along with their 95% CIs for each study.

A detailed description of the interventions for the experimental and control groups of these studies is reported in Table 3.

#### 3.7.1. Comparison Assessed with the Berg Balance Scale

The studies by Kotov et al. [27], Louie et al. [31] and Wall et al. [29] were considered. These studies, conducted on a subacute population, compared the efficacy of o-RAGT with and without CT or CT on balance. The three interventions were different according to the o-RAGT device (ExoAtlet, Ekso GT and HAL), the frequency (from 3 up to 5 sessions per week) and the total duration of interventions (from 2 up to 8 weeks). The meta-analysis revealed no statistically significant results (*p* = 1.00; mean difference = −0.01, 95% confidence interval (CI) = −3.24, 3.26) (see Figure 4a). Considering the estimated minimal clinically important difference (MCID) of the BBS score for individuals with subacute stroke [33], the results of this quantitative analysis did not reveal any considerable clinical improvement worthwhile for the sample as the lower limit in the effect size was lower than the MCID. Lastly, the I^2^ value was equal to 0% for heterogeneity.

#### 3.7.2. Comparison Assessed with the Barthel Index

The studies by Kotov et al. [28], Rojek et al. [30] and Wall et al. [29] were considered. These studies compared the efficacy of o-RAGT with and without CT and CT on ADL. The three interventions were different according to the o-RAGT device (ExoAtlet, Ekso GT and HAL), the frequency (from 3 up to 5 sessions per week) and the total duration of interventions (from 2 up to 8 weeks). The meta-analysis revealed no statistically significant results (*p* = 0.76; mean difference = −3.35, 95% confidence interval (CI) = −24.46, 17.75) (see Figure 4b). The I^2^ value was equal to 85% for heterogeneity.

#### 3.7.3. Subgroup Analysis

Despite individual participants’ data has been requested to the authors, it was not possible to conduct subgroups analysis due to the not availability of the data.

## 4. Discussion

The primary aim of this meta-analysis was to investigate the efficacy of o-RAGT on balance and secondarily on ADL in stroke individuals. Balance recovery is considered one of the main goals of neurorehabilitation programs [8], as balance impairments drastically limit the ADL of individuals with stroke [34]. Modern concepts favor task-specific repetitive rehabilitation approaches [35], with high intensity [36] and early multisensory stimulation [37]. In stroke rehabilitation, good outcomes are strongly associated with a high degree of motivation, participation [38,39,40] and good cognitive function, especially attention [41]. A prerequisite for learning is the recognition of the discrepancy between the actual and expected outcomes during error-driven learning [42]. For all the devices included in this meta-analysis, according to the features of each exoskeleton, the participants received information about their performances provided by the device itself. Furthermore, it has been reported that the task-specific repetitive practice provided by o-RAGT may offer more realistic task-specific and goal-oriented overground walking practice than treadmill-based devices [43], enabling the patients to experience increased proprioceptive input when compared with the stationary treadmill training [44]. These factors may suggest that o-RAGT devices allow an increment in patient motivation, participation and attention. In recent years, various powered overground exoskeletons have been commercially developed to assist and allow overground walking [31,45]. o-RAGT efficacy on balance function was partially considered in a single meta-analysis that did not include only full leg EXOs and in which the devices were mostly associated with BWS. Given this lack of information about o-RAGT usage efficacy on balance and ADL in stroke individuals, a systematic review and meta-analysis were conducted on these topics.

In the nine included studies, four different o-RAGT devices were addressed: EksoGT (n = 3 [28,30,31]), HAL (n = 3 [24,25,29]), ExoAtlet (n = 2 [26,27]) and H2 (n = 1 [32]). Only two studies [28,30] out of nine had as primary aim the measurement of the o-RAGT efficacy on balance, proposing an EksoGT device training associated with CT. Instead, the other studies focused primarily on gait recovery. These data highlight the necessity of focusing devoted studies on assessing o-RAGT efficacy on balance function.

The average level of methodological quality across studies was moderate according to the D&B tool (see Table 1). Although a control group was present in most of the studies, the samples were low, the follow-up examinations were rare, and the statistical analysis scarcely focused to understand demographic and clinical features’ influence on recovery. This moderate level of methodological quality seems to be in contrast with the growing recent interest in o-RAGT. A recent study [46], aimed at assessing the quality of the systematic reviews based on o-RAGT devices usage in neurological disorders, highlighted the poor methodological and reporting quality of these studies. This evidence, in line with the results of this review, emphasizes the need to conduct studies with a higher methodological quality on the stroke population.

The analysis of epidemiological data in the nine studies (see Table 2) showed that the number of enrolled individuals was heterogeneous and, even if six RCT [26,27,28,29,30,31] were included, the total number of enrolled individuals was low. In addition, the studies included samples with a mean age above 44 years. The incidence of stroke rapidly increases with age, doubling for each decade after age 55 [47] and over 70% of all strokes occur at the age of 65 years [48]. Only 5/10% of acute cerebrovascular events occur in people younger than 45 years of age [38]. In this subpopulation of young adults, the motor outcome of cerebral damage is better than in older patients [49,50]. Therefore, the effect of o-RAGT on balance in older subjects deserves future research.

Regarding the time since stroke, in the literature, it is reported that a greater potential for improvement is shown by individuals with stroke in the subacute phase of recovery when receiving neurorehabilitation (both CT [51] or t-RAGT [15]) than individuals in the chronic phase. A statistically significant balance function improvement was obtained in only three out of the nine studies included [26,28,30] (see Table 4): one of them focused on a population with subacute stroke [26], one on a population in the chronic phase [28] and one on both the subacute and chronic phases [30]. The results of these three studies are controversial: Calabrò et al. [28] reported improvements only in the EksoGT group after training, Kotov et al. [26] showed significant improvements in both the ExoAtlet and Cy-E groups and, on the other hand, Rojek et al. [30] showed improvement in the deviation X-OE and Y-OE conditions only for the EksoGT group and in the Y-CE condition only for the CT group. Moreover, Kotov et al. [27] did not report a significant difference in the post-training evaluation for both groups, but showed a greater improvement in the ExoAtlet group when comparing post–pre training delta values. Additionally, experimental groups changes were often greater than those of the control groups in the outcome change comparisons. However, it is not currently clear from the results of this review if individuals with stroke could benefit more from o-RAGT than CT or other interventions. Although previous studies showed greater benefits for individuals in the subacute phase after t-RAGT [15], there is not enough evidence to confirm it for o-RAGT. Furthermore, the balance improvements obtained in the study enrolling a population with chronic stroke [28] open the door to the need for more research into the effects of o-RAGT also in this population.

As regards the ability to walk independently, previous evidence [51,52,53] reported that individuals affected more severely can have greater t-RAGT benefits than those who are less affected. Considering the small number of individuals included in the nine studies, this issue remains open because it is not possible to confirm whether o-RAGT can also be useful for individuals affected more severely.

The results of this systematic review and meta-analysis indicate that BBS was the most used tool for functional balance assessment in the stroke population. In addition to clinical scales, instrumental assessments can also measure the balance function in terms of CoP data changes. A previous study [54] suggested that combining quantitative CoP evaluation and clinical assessment, whenever possible, would enhance the comprehension of balance impairments and disabilities in individuals with stroke. However, the instrumental assessments of balance are still underutilized [30], even if these may be more objective than clinical scales. A total of six studies [24,25,28,29,31,32] measured the o-RAGT efficacy on balance only by clinical scales, two studies [26,27] through both clinical scales and instrumental assessments, and lastly a single study [30] conducted instrumental assessment alone (see Figure 4a). This evidence confirmed that, although the need for both clinical and instrumental assessments has been suggested in the last decade, this is not the case in clinical practice. It is interesting to note that the three studies [26,27,30] that selected instrumental balance assessments were published between 2020 and 2021. This may suggest an increased interest in instrumental evaluation or the greater availability of the technological devices to carry them out. Furthermore, significant improvements were highlighted in two studies [26,27] in which instrumental analysis was associated with the administration of clinical scales for the comparisons between intervention groups or pre- vs. post-training. These data are in line with that of Lin et al. [55], who suggested that it is recommended that clinicians consider the use of both clinical balance scales and instrumental balance measurements when assessing stroke patients to improve the accuracy of assessments, leading to a better individualized treatment plan.

In the six RCTs included, the control groups underwent CT, only with the exception of Kotow et al. [26], in which the control group received Cy-E training. Only for three RCTs, significant balance improvements were reported after experimental or control trainings in the comparison of pre- vs. post-training. For the two RCTs aimed at comparing EksoGT device plus CT versus CT alone, balance improved only for the experimental groups. Instead, in the RCT [26] in which the ExoAtlet device was compared with Cy-E training, balance improved after both the control and experimental trainings. These balance enhancements were reported for the intervention periods of between 2 and 8 weeks. These interventions were carried out as intensive training with a frequency of 5 days per week. Only a single pilot study [24], with the same frequency, did not report significant improvements after training. These findings might suggest that daily training could potentially allow to balance enhancements. It is intriguing that, taking into account the comparison between the experimental (EksoGT and ExoAtlet devices) and control groups (CT or Cy-E), significant differences were reported after training. These differences were in favor of the groups that received o-RAGT training, alone or in association with CT, suggesting a higher impact of o-RAGT efficacy on balance after stroke even if both types of training improved balance performances.

Despite the significant results obtained from these studies in favor of EksoGT or ExoAtlet devices, the meta-analysis performed on the BBS score after training suggested that o-RAGT did not increase balance more than CT and that the heterogeneity of the data might not be important (see Figure 4). This contradictory finding should be evaluated considering that, among the three RCTs included in the meta-analysis, only one corresponds to those for which a greater influence of the ExoAtlet device on balance was found compared to the CT. Moreover, the reduced number of included studies with available information and the heterogeneity of the stroke population included represent limitations in the interpretation of the results of the meta-analysis. In addition, it was not possible to conduct a subgroups analysis due to the missing information of individual participants. Future studies including these missing data are needed to better investigate the o-RAGT efficacy on balance after stroke.

Furthermore, in the area of RAGT, to date, there are no studies comparing the effects of treatments with end-effector versus o-RAGT devices. This lack of data in the literature leaves open a very interesting point of discussion. In fact, the main difference between end-effector and overground devices is the origin of motion. In the case of end-effectors, the motion is generated by the device starting from the periphery, while for the other devices, it is the exoskeleton itself that generates the motion in all the joints. This difference is intriguing because most patients with stroke experience ankle–foot disability [56]. Following stroke, foot deformity, altered plantar sensory inputs, reduced ankle proprioception, altered motor control or toe clawing have all been observed, identifying a relationship between these impairments and balance impairments [57]. Specific foot and ankle impairments may also negatively contribute to perceptions of physical appearance and self-esteem as well as the quality of life being severely affected in stroke survivors, specifically in walking independently, due to the reduced peak of the ankle dorsiflexion angle in the paretic leg [58]. Future studies aimed at comparing different type of s-RAGT vs. o-RAGT may shed light on this aspect that has not yet been investigated.

The secondary aim of this systematic review and meta-analysis was to analyze o-RAGT usage efficacy on ADL. Only four RCTs [26,27,29,30] out of the nine studies addressed this issue, selecting mainly the BI [26,27,29,30], which is considered an adequate tool for assessing the functional status of patients after stroke and it is a good indicator of the therapy efficacy [59,60,61]. Only two [26,30] out of four RCTs reported significant changes (see Figure 4b) in the comparison of pre- vs. post-training. In both studies, a significant improvement in ADL was reported either in the experimental or control groups. In Kotov et al. [26], the improvements were greater in the ExoAtlet group when compared versus Cy-E and, in Rojek et al. [30], the group that received EksoGT training associated with the CT had a BI score lower than that of the control group (CT) at the baseline assessment, but the improvement int the experimental group was stronger than that in the CT. It is interesting that, in both studies, in addition to ADL enhancement, significant improvements in balance were also reported. Further investigations are needed to assess if a direct relationship between balance and ADL improvements due to o-RAGT is present in the stroke population [33]. The meta-analysis was conducted on three out of four RCTs that were focused on ADL evaluation, of which only one reported a significant change due to the EksoGT device plus CT. The results of the meta-analysis suggest that o-RAGT does not increase the BI score more than CT after o-RAGT. Contrary to the BBS meta-analysis results, the results of the meta-analysis conducted on BI indicate the presence of considerable heterogeneity across the data.

There was no follow-up examination in any of the studies reported above with significant changes in balance function or ADL. Thus, it is not possible to understand whether the o-RAGTs effects were maintained in the long term. In fact, only three out of the nine included studies reported follow-up assessments [25,29,31], but they did not report significant changes either after training or at the follow-up assessments.

Considering the above data and the growing interest in o-RAGT devices in the stroke rehabilitation framework, in the absence of adverse events due to o-RAGTs, it is necessary to conduct good quality RCTs with uniform control groups to better understand the efficacy of o-RAGT devices for the recovery of balance and ADL after stroke. Moreover, future studies should focus on analyzing the o-RAGT efficacy on balance and ADL improvement, considering clinical and demographic factors, such as time onset (subacute or chronic), disease severity, age and gender. Finally, information regarding the effects of different training dosages and different frequencies of training should be addressed.

The search string and the inclusion of only English-language studies may have resulted in missing additional studies available in the literature. In addition, this meta-analysis is indirectly limited by the reduced number of included studies, the small heterogeneous number of participants with variable dosage and type of interventions and the lack of a uniform presence of follow-up assessments. Furthermore, the meta-analysis conducted on the BBS and BI clinical scales included only three studies each and the lack of single-participant data did not allow us to conduct a meta-analysis by subgroups.

## 5. Conclusions

The current review provides information on the efficacy of o-RAGT on the balance function and ADL in stroke survivors. Although different studies reported positive effects, improvements due to o-RAGT on balance and ADL were not greater than those obtained by means of other rehabilitation therapies. The low methodological quality, heterogeneity and the small number of the studies included does not allow general conclusions to be reached about the usefulness of o-RAGT on balance and ADL in patients with stroke. Further well-designed RCTs are needed.

## Figures and Tables

**Figure 1 brainsci-12-00713-f001:**
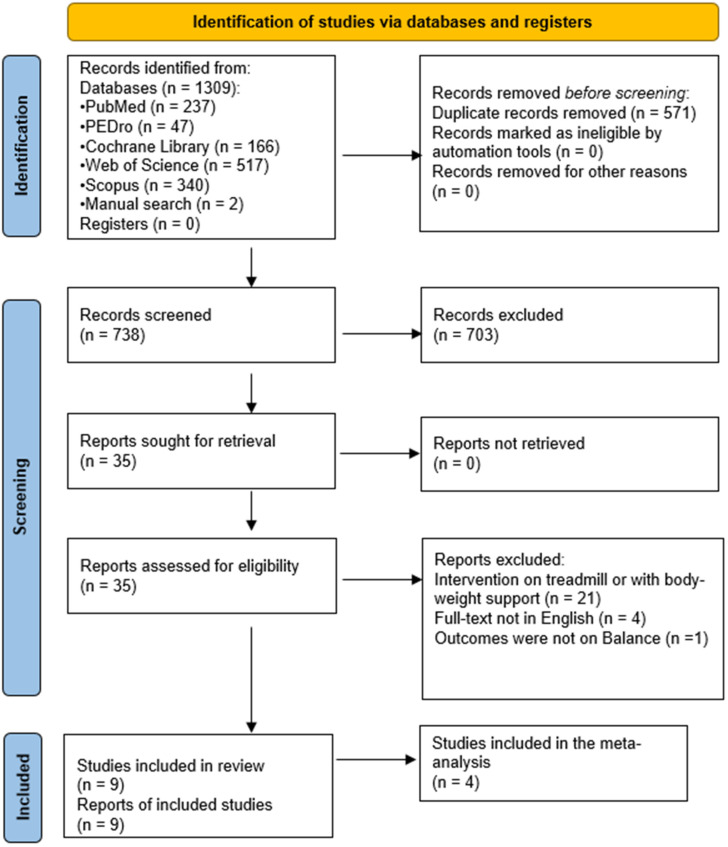
PRISMA flow diagram [19] of the study selection process.

**Figure 2 brainsci-12-00713-f002:**
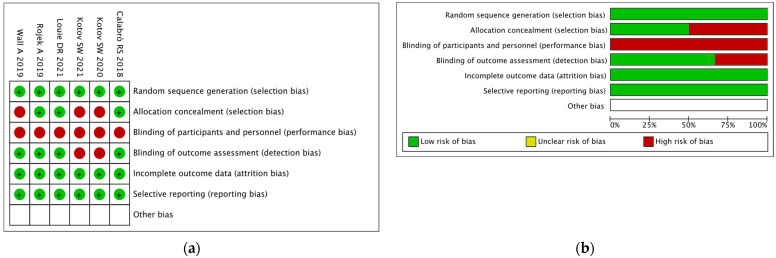
Cochrane risk of bias tool: (**a**) review authors’ judgements about each risk of bias item for each included study; (**b**) review authors’ judgements about each risk of bias item presented as percentages across all included studies.

**Figure 3 brainsci-12-00713-f003:**
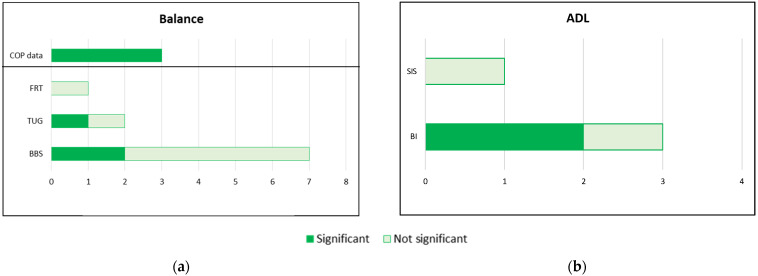
(**a**) Number of studies adopting instrumental or clinical scales for the balance function; (**b**) Number of studies adopting instrumental or clinical scales for ADL (BI: Barthel Index; BBS: Berg Balance Scale; COP: Center of Pressure; FRT: Functional Reach Test; SIS: Stroke Impact Scale; TUG: Timed Up and Go).

**Figure 4 brainsci-12-00713-f004:**
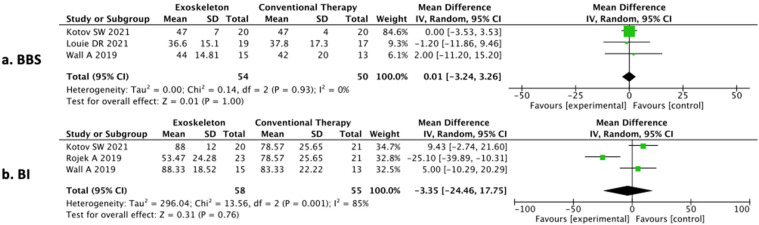
(**a**) Comparison of o-RAGT with or without CT vs. CT data per the BBS; (**b**) Comparison of o-RAGT vs. CT data per the BI (BBS: Berg Balance Scale; BI: Barthel Index; CI: Confidence interval; SD: Standard deviation).

**Table 1 brainsci-12-00713-t001:** Downs and Black (D&B) tool sub-sections and total scores are reported for each study in decreasing order (RCT: Randomized Controlled Trial).

Study	Study Design	Reporting	External Validity	Internal Validity		Power	Total Score
				Bias	Confounding		
Louie DR et al., 2021 [31]	RCT	11	3	6	5	1	26
Calabrò RS et al., 2018 [28]	RCT	10	3	5	4	1	23
Wall A et al., 2019 [29]	RCT	10	2	5	6	0	23
Rojek A et al., 2019 [30]	RCT	8	3	5	3	0	19
Kotov SW et al., 2020 [26]	RCT	9	1	4	2	0	16
Kotov SW et al., 2021 [27]	RCT	7	1	4	2	0	14
Mizukami M et al., [24] 2017	Pilot study	10	1	3	0	0	14
Yoshimoto T et al., [25] 2016	Case Report	8	1	2	1	0	12
Bortole M et al., 2015. [32]	Pilot study	8	0	2	0	0	10

**Table 2 brainsci-12-00713-t002:** Demographic and clinical features data of participants classified according to D&B tool total score (CTRL: control; D: dependent ambulator; d: days; EXP: experimental; F: female; h: hemorrhagic; I: independent ambulator; i: ischemic; l: left; M: male; m: months; NR: not reported; r: right; SD: standard deviation; TSS: time since stroke).

Study	Individuals Enrolled	Individuals Completing the Trial	Demographic Features		Clinical Features				
			Gender	Age: Mean ± SD	TSS	Recovery Phase	Stroke Type	Hemiparesis Side	Walking Independence
Louie DR et al., 2021 [31]	36	36	M: 26, F: 10	EXP group: 59.6 ± 15.8	EXP group: 36.7 ± 19.0 d	Subacute	i: 25, h: 11	l: 21, r: 15	D: 36
				CTRL group: 55.3 ± 10.6	CTRL group: 40.9 ± 19.8 d				
Calabrò RS et al., 2018 [28]	40	40	M: 23, F: 17	EXP group: 69 ± 4	EXP group: 10 ± 3 m	Chronic	i: 40	l: 23, r: 17	D + I: 40
				CTRL group: 67 ± 6	CTRL group: 11 ± 3 m				
Rojek A et al., 2019 [30]	60	44	M: 25, F: 19	69 ± 7	4–12 m	Subacute & Chronic	i: 44	l: 24, r: 20	NR
Wall A et al., 2019 [29]	34	28	M: 23, F: 5	53 ± 12	NR	Subacute	i: 16, h: 12	l: 20, r: 8	D: 28
Mizukami M et al., 2017 [24]	10	10	M: 5, F: 3	58.6 ± 16.91	132.6 ± 18.6 d	Subacute	i: 3, h: 5	l: 3, r: 5	D: 7, I: 1
Yoshimoto T et al., 2016 [25]	1	1	F: 1	~60	57 m	Chronic	h: 1	l: 1	I: 1
Kotov SW et al., 2020 [26]	47	41	M: 28, F: 19	62.9 ± 11.0	2.2 ± 1.2 m	Subacute	i: 47	l: 29, r: 18	D + I: 47
Kotov SW et al., 2021 [27]	42	42	NR	61.2 ± 9.3	NR	Subacute	i: 42	NR	D + I: 42
Bortole M et al., 2015 [32]	3	3	M: 3	48.7 ± 8.1	25.7 ± 29.8 m	Chronic	NR	l: 3	NR

**Table 3 brainsci-12-00713-t003:** Intervention data are classified according to the type of device and D&B tool total scores. (CT: Conventional Therapy; Cy-E: Cyclo-ergometer; HAL: Hybrid Assistive Limb; N: not executed; m: months; NR: not reported; w: weeks).

Study	Total Number of Session for Each Group	Total Duration for Each Group	Experimental Group			Control group		
			Intervention	Single Session Duration (Minutes)	Frequency (Times per Week)	Intervention	Single Session Duration (Minutes)	Frequency (Times per Week)
Louie DR et al., 2021 [31]	40	8 w	EksoGT + CT	EksoGT: 45	3	CT	60	4–5
				CT: 15				
				CT: 60	1–2			
Calabrò RS et al., 2018 [28]	40	8 w	EksoGT + CT	Exo: 45	5	CT	105	5
				CT: 60				
Rojek A et al., 2019 [30]	20	4 w	EksoGT + CT	Exo: 45	5	CT	105	5
				CT: 60				
Wall A et al., 2019 [29]	16	4 w	HAL + CT	NR	4	CT	NR	4
Mizukami M et al., 2017 [24]	20–25	5 w	HAL + CT	HAL: 20	5	N		
				CT: 40				
Yoshimoto T et al., 2016 [25]	NR	24 w	CT	40	NR	N		
			CT + HAL	60	1			
			CT	40	NR			
Kotov SW et al., 2020 [26]	10	2 w	ExoAtlet	10–30	5	Cy-E	10–30	5
Kotov SW et al., 2021 [27]	10	2 w	ExoAtlet + CT	Exo: 10–30	5	CT	20–40	5
				CT: 20–40				
Bortole M et al., 2015 [32]	12	4 w	H2	30	3	N		

**Table 4 brainsci-12-00713-t004:** Results of the balance and ADL clinical and instrumental outcome measures.

Study (Device Name)	Main Goal	Clinical Assessment	Instrumental Assessment	Clinical Scale Results	Instrumental Assessment Results
Louie DR et al., 2021 [31] (Ekso GT)	To compare walking independence of non-ambulatory patients using an exoskeleton versus patients who received standard physical therapy. The secondary objective was to evaluate the effect of exoskeleton-based physical therapy on additional walking and mobility outcomes (e.g., speed), leg motor impairment, balance, cognition, post-stroke depression, and quality of life.	FAC, 5 MWT, 6 MWT, number of days to achieve unassisted ambulation, FMA-LE, BBS, PHQ, MoCA, SF-36	N	post training: Exo vs. CT BBS: Exo > CT SF-36 physical: Exo > CT SF-36 mental: Exo < CT FU: Exo vs. CT BBS: Exo < CT SF-36 physical: Exo > CT SF-36 mental: Exo < CT	N
Calabrò RS et al., 2018 [28] (Ekso GT)	To obtain an improvement in lower limb gait and balance at the end of the training getting the MCID for the 10 MWT, RMI, and TUG scales.	10 MWT, RMI, TUG	EMG data, EEG data, Gait analysis data (spatio-temporal parameters)	pre vs. post training: TUG: Exo group ↓*, OGT group ↓ Δ Exo vs. Δ OGT (Δ = post - pre training): TUG: Exo < OGT *	N
Rojek A et al., 2019 [30] (Ekso GT)	To evaluate the effects of Ekso GT exoskeleton-assisted gait training on balance, load distribution, and functional status of patients after ischemic stroke.	BI, RMI	COP data OE and CE: L, V, length of minor axis, length of major axis, ellipse angle, deviation X, deviation Y; load distribution: total load, forefoot load, backfoot load	pre vs. post training: BI: Exo group ↑ ***, CT group ↑ * pre training: Exo vs. CT BI exo < CT *** post training: Exo vs. CT BI exo < CT *	pre vs. post training: L-OE: Exo ↓, CT ↑; L-CE: Exo ↓, CT ↑ V-OE: Exo ↓, CT ↑; V-CE: Exo ↑, CT ↑ Length of minor axis-OE: Exo ↓, CT ↑; Length of minor axis-CE: Exo ↓, CT ↑ Length of major axis-OE: Exo ↓, CT ↑; Length of major axis-CE: Exo ↓, CT ↓ deviation X-OE: Exo ↓*, CT ↓: deviation X-CE: Exo ↓, CT: ↑ deviation Y-OE: Exo ↓ *, CT ↓; deviation Y-CE: Exo ↓, CT ↓ * pre training: Exo vs. CT L-OE: Exo < CT; L-CE: Exo > CT V-OE: Exo < CT; V-CE: Exo > CT Length of minor axis-OE: Exo > CT; Length of minor axis-CE: Exo > CT Length of major axis-OE: Exo > CT; Length of major axis-CE: Exo > CT deviation X-OE: Exo > CT; deviation X-CE: Exo > CT deviation Y-OE: Exo > CT **; deviation Y-CE: Exo > CT post training: Exo vs. CT L-OE: Exo < CT *; L-CE: Exo < CT V-OE: Exo < CT *; V-CE: Exo < CT Length of minor axis-OE: Exo < CT; Length of minor axis-CE: Exo > CT Length of major axis-OE: Exo = CT; Length of major axis-CE: Exo > CT deviation X-OE: Exo < CT; deviation X-CE: Exo < CT deviation Y-OE: Exo > CT *; deviation Y-CE: Exo > CT *
Wall A et al., 2019 [29] (HAL)	To explore long-term effects of HAL exoskeleton usage compared to conventional gait training in the subacute stage after stroke, regarding self-perceived functioning, disability and recovery and factors associated with self-perceived recovery.	NIHSS, SIS: strength (domain 1), ADL (domain 5), mobility (domain 6), and participation (domain 8), BBS	N	pre vs. FU: BBS: ↑ both groups BI: ↑ both groups Δ Exo vs. Δ CT (Δ = baseline - FU): BBS Exo > CT BI Exo < CT Exo vs. CT (FU): SIS ADL: Exo = CT Correlation between self-perceived mobility SIS and BBS	N
Mizukami M et al., 2017 [24] (HAL)	To determine whether gait training with a hybrid assistive limb (HAL) device was safe and could increase functional mobility and gait ability in subacute stroke patients.	MWS, SWS, 2 MWT, FAC, FMA, BBS, PCI	N	pre vs. post training: BBS: ↑	N
Yoshimoto T et al., 2016 [25] (HAL)	To investigate the accumulated and sustained effects of Hybrid Assistive Limb gait training in a subject with chronic stroke.	10 MWT, number of steps and cadence, TUG, FRT, 2 ST, BBS	N	pre vs. post CT period and pre vs. post HAL period: TUG ↓, BBS ↑, FRT ↑ post HAL-FU: TUG ↑, BBS ↑, FRT ↓	N
Kotov SW et al., 2020 [26] (ExoAtlet)	To compare the effectiveness of restoration of walking function in patients with ischemic stroke using a lower limb exoskeleton and an active-passive pedal bicycle trainer.	MRC, MAS, BBS, HAI, 10 MWT, Rankin scale, BI	COP data: L, surface area of the statokinesiogram, energy consumption during Romberg Test with OE or CE; Biomechanichal and EMG data during walking.	pre vs. post training: BBS: Exo ↑ ***, Cy-E ↑ *** BI: Exo ↑ ***, Cy-E ↑ *** Δ Exo vs. Δ Moto (Δ = post - pre training): BBS: Exo > Cy-E * BI: Exo > Cy-E *	pre vs. post training: L-OE: Exo ↓ ***, Cy-E ↓ *** L-CE: Exo ↓ ***, Cy-E ↓ * S-OE: Exo ↓ ***, Cy-E ↓ *** S-CE: Exo ↓ *, Cy-E ↓ *** Ei-OE: Exo ↓ ***, Cy-E ↓ *** Ei-CE: Exo ↓ ***, Cy-E ↓ *** Δ Exo vs. Δ Moto (Δ = post - pre training): L-OE: Exo > Cy-E ***
Kotov SW et al., 2021 [27] (ExoAtlet)	To evaluate the effectiveness of ExoAtlet usage in restoring the functional and motor activity, including the walking function, in patients after ischemic stroke in the middle cerebral artery, compared with the traditional methods of rehabilitation.	MRC, Rankin scale, BI, HAI, BBS, 10 MWT	COP data: L, surface area of the statokinesiogram, energy consumption during Romberg Test with OE or CE	pre training: Exo vs. CT BBS: Exo = CT BI: Exo = CT post training: Exo vs. CT BBS: Exo > CT BI: Exo > CT Δ Exo vs. Δ CT (Δ = post - pre training): BBS: Exo > CT * BI: Exo > CT	pre training: Exo vs. CT COP data: L-OE: Exo < CT; L-CE: Exo < CT S-OE: Exo < CT; S-CE: Exo < CT Ei-OE: Exo < CT; Ei-CE: Exo < CT * post training: Exo vs. CT COP data: L-OE: Exo < CT ***; L-CE: Exo < CT * S-OE: Exo < CT ***; S-CE: Exo < CT ** Ei-OE: Exo < CT ***; Ei-CE: Exo < CT ***
Bortole M et al., 2015 [32] (H2)	To demonstrate safety and usability of the H2 robotic exoskeleton in post-stroke hemiparetic patients in a rehabilitation framework.	BBS, BI, FGI, FMA-LE, TUG, 6 MWT	N	pre vs. post training: BBS: patient 1 ↑, patient 2 e 3 = TUG: patient 1 ↑, patient 2 e 3 ↓ BI ADL: patient 1 e 3 = patient 2 ↑	N

The type of comparison is specified within the cells. In the case of an increase in the data between evaluation time points, “↑” is reported, while in the case of a reduction in the data between evaluation time points, “↓” is reported. In the case of comparison between groups or between different groups, “ >” or “< ” are used. If no changes are reported, “=” is used. If the authors of the studies identified significant data variations, results are reported in bold characters. Asterisks indicate statistically significant variations (* *p* < 0.05; ** *p* < 0.01; *** *p* < 0.001). If differences between evaluation time points are compared, “Δ” is used. (2 MWT: 2 min walk test; 5 MWT: 5 min walk test; 6 MWT: 6 min walk test; 10 MWT: 10-m walk test; 2 ST: 2-step test; ADL: Activities of daily living; BBS: Berg Balance Scale; BI: Barthel Index; CE: Closed eyes; COP: Center Of Pressure; CT: Conventional Therapy; Cy-E: Cyclo-ergometer; DS: Digit Span subset; DST: WAIS-R digit symbol test; EEG: Electroencephalogram; Ei: Energy index (COP data); EMG: Electromyography; Exo: Exoskeleton; FAC: Functional Ambulatory Category; FGI: Functional Gait Index; FMA-LE: Fugl-Meyer’s assessment of motor recovery (lower extremity); FRT: Functional Reach Test; HAI: Hauser Ambulation Index; HAL: Hybrid Assistive Limb; L: Length (COP data); MAS: Modified Ashworth Scale; MMSE: Mini-Mental State Examination; MoCA: Montreal Cognitive Assessment; MRC: Medical Research Council Scale; MWS: Maximum Walking Speed; NIHSS: National Institutes of Health Stroke Scale; OE: open eyes; PCI: Physiological Cost Index; PHQ: Patient health questionnaire; RCT: Randomized Controlled Trial; RMI: Rivermead Medical Index; S: Surface sway (COP data); SF-36: Medical Outcomes Short-Form 36; SIS: Stroke Impact Scale; ST: Stroop Test; SWS: Self-selected Walking Speed; TMT: Trail Making Test; TUG: Timed Up and Go test; V: Velocity speed (COP data); WAIS-R: Wechsler Adult Intelligence Scale-Revised).

## Data Availability

Not applicable.

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
