# Peer review of "Efficacy of Overground Robotic Gait Training on Balance in Stroke Survivors: A Systematic Review and Meta-Analysis"

_brainsci, 2022, doi:10.3390/brainsci12060713_

Round 1

Reviewer 1 Report

The manuscript investigates effects of overground robotic gait training on balance in stroke survivors through a systematic review and meta-analysis. The manuscript provides some important results regarding stroke rehabilitation. The manuscript is extremely long and hard to follow. It needs to be fully revised. It should be rewritten in shorter and more informative form. There is an exaggeration in the description of tables and figures.

However, I see that the manuscript needs to be fully revised and rewritten. Please find my following comments:
1- Title: I suggest to be effectiveness rather that effects
2- Please clarify more the difference between overground robotic gait training, and other types of robotics such as robot assisted gait training. You can use the required images.
3- The inclusion criteria for meta-analysis is not clear
4- The number of the included studies in the PRISMA chart is missing
5- Please be more aware on the clinical and demographic data that are more related to your primary outcomes such as time since stroke.   
6- You can mention that the BBS was not improve statistically or clinically by using the minimal clinical importance difference as the lower limit in the effect size was lesser than the minimal clinical importance difference. doi: 10.1016/j.apmr.2018.03.0257- Recently, a number of good meta-analyses has been published about the different types of robotics on motor outcomes in neurorehabilitation. Please read them carefully that can help you in the revising of the manuscript

Author Response

Dear Reviewers, thanks for all your comments and contributes and for the possibility to improve the manuscript. Please find in red in this document the answer to your comments.

Reviewer 1

The manuscript investigates effects of overground robotic gait training on balance in stroke survivors through a systematic review and meta-analysis. The manuscript provides some important results regarding stroke rehabilitation. The manuscript is extremely long and hard to follow. It needs to be fully revised. It should be rewritten in shorter and more informative form. There is an exaggeration in the description of tables and figures.
However, I see that the manuscript needs to be fully revised and rewritten. Please find my following comments: 
1- Title: I suggest to be effectiveness rather that effects 
Thank you for your comment. We agree that effects is too general. We have modified according to your suggestion effects with “Efficacy”. Consequently, we modified the word in the whole text.

Sorry if we did not accept your suggestion “Effectiveness”. The reason was that term effectiveness in neurorehabilitation is from several year associated to this concept of Shah and Vanclay: .Effectiveness reflects the proportion of potential improvement that was achieved during rehabilitation, calculated as [(discharge score − initial score)/(maximum score − initial score)] × 100. Thus, if a patient achieved the highest possible score after rehabilitation, the effectiveness was 100%.
REF: Shah S, Vanclay F, Cooper B. Efficiency, effectiveness, and duration of stroke rehabilitation. Stroke. 1990 Feb;21(2):241-6. doi: 10.1161/01.str.21.2.241. PMID: 2305399.
2- Please clarify more the difference between overground robotic gait training, and other types of robotics such as robot assisted gait training. You can use the required images. 
Thank you for the possibility to better explain the differences between the different device types. We have modified and better ordered this section to make it easier to read.
3- The inclusion criteria for meta-analysis is not clear.
Dear Reviewer, thanks for the possibility to better clarify this topic. We modified this section adding more information on the inclusion criteria for meta-analysis, the section was improved as follow: A meta-analysis was performed to compare the post-intervention changes between the experimental group (o-RAGT with or without CT) and the control group when at least 3 RCTs [15] were available that provided the same treatment to the control group and measured changes using the same clinical scale for both assessments, balance and ADL. The studies were grouped according to the outcome measure and the standardized mean differences (MDs) were calculated, together with their 95% confidence intervals (CIs). An I2 value > 40% was considered as the threshold for statistical heterogeneity [24].”
4- The number of the included studies in the PRISMA chart is missing 
Thank you for pointing out this missing information. We added the number of the included studies in the PRISMA chart.
5- Please be more aware on the clinical and demographic data that are more related to your primary outcomes such as time since stroke.   

We thank the reviewer for raising this important point regarding the clinical implication. We improved the discussion in light of the reviewer's consideration. In particular sentence regarding subject severity was improved as follow: “As regards the ability to walk independently, previous evidence [51,52] reported that more severe individuals can have greater RAGT benefits than less affected ones. Considering the low number of individuals included in the 9 studies, this issue remains open because it is not possible to confirm whether o-RAGT may be more useful for more severe individuals.”

In addition, sentence regarding the age were improved as follow: “Therefore, the effect of o-RAGT on balance in older subjects deserves future researches.”

Furthermore, due to the reduced number of subject included, no conclusion can be made on clinical and demographic data. So we have added the following sentence at the end of the discussion: " Moreover, future studies should focus on analyzing the o-RAGT effectiveness on balance and ADL improvement, considering clinical and demographic factors such as time onset (subacute or chronic), disease severity, age and gender. "

6- You can mention that the BBS was not improve statistically or clinically by using the minimal clinical importance difference as the lower limit in the effect size was lesser than the minimal clinical importance difference. doi: 10.1016/j.apmr.2018.03.025
Thank you for your suggestion. We added that the meta-analysis results did not reveal any considerable clinical improvement according to the MCID, this part was integrated in the text as follow: “Considering the estimated minimal clinically important difference (MCID) of the BBS score for individuals with subacute stroke, results of this quantitative analysis did not reveal any considerable clinical improvement worthwhile for the sample as the lower limit in the effect size was lesser than the MCID.” In addition, a new reference was included (Tamura S, Miyata K, Kobayashi S, Takeda R, Iwamoto H. The minimal clinically important difference in Berg Balance Scale scores among patients with early subacute stroke: a multicenter, retrospective, observational study. Top Stroke Rehabil. 2021 Jun 25:1-7. doi: 10.1080/10749357.2021.1943800)
7- Recently, a number of good meta-analyses has been published about the different types of robotics on motor outcomes in neurorehabilitation. Please read them carefully that can help you in the revising of the manuscript 
Thank you for your indication. As you suggested we carefully read the latest meta-analysis on robotics devices and according to them we have revised the manuscript.

Reviewer 2 Report

Dear authors 
It was with great interest that I read your systematic review of over-ground RAGT to affect change in balance after stroke . Indeed over-ground systems require initiation of movement and engage the vestibular system and additional postural control pathways in a way over-treadmill walking doesn't. There is good evidence of an altered neurological basis of treadmill walking at muscle (EMG) and brain (EEG)  systems so I was interested to read your review.

Your introduction sets the scene well and creates a good case for conducting your review.

You registered your review and have conducted it with good scientific rigour in keeping with the guidelines. The search is still current to Nov 2021.

The search terms as presented appear a little basic and do not include controlled vocabularly terms in databases. The single use of balance for your outcome is limited as it may miss for example equilibrium, centre of gravity/mass, postural control etc. and you did not search for "electromechanically assisted gait" in your robotic search string. You will need to identify these as study limitation at the end along with the decision to look for only english language studies. You may have missed additional studies because of these choices.

You have chosen to report 3 quality assessment tools and this is excessive I think. I would be inclined to report the D&B for all study types and the cochrane ROB for RCTs. 

In your data synthesis section please clarify that it is a minimum of 3 RCTs with the same outcome measures for meta-analysis.

The next area of confusion is what you are labelling RCTs you say there are 3 and then include 6 in the Cochrane ROB assessment and use these in the meta-analysis. Looking at all 6 of these studies -they should all be classified as RCTs- this would make this section a lot cleaner and allow you -in the narrative synthesis -to discuss and report by level of evidence, RCTs first , then pilot studies (identifying them by methodology better- eg prospective cohort study/ non-randomised case control study etc as appropriate), then case report. In table one you label the Kotov studies as clinical prospective studies yet they have 2 groups that are randomised and you use a ROB tool for RCTs later.

I would delete table 2 and PEDro rating here and in figure 2 identify ROB tool for RCTs.

In 3.6 outcome measure- you need to provide detail with respect to what instrumentation measures were used for balance and again provide greater synthesis by study type rather than reporting each study seperately .

In figure 3(b) please remove the decimal place for the discrete variable of number of studies. Add a legend for COP; FRT etc so the figure can stand alone in its interpretation.

In your meta-analysis figure 4 please identify on the label that it is a comparison of 0-RAGT with or without conventional therapy vs conventional therapy. I would advise you to do a further sensitivity analysis  by removing the Kotov study and looking at o-RAGT with CT vs CT alone, reporting the analysis results on the 2 pooled studies and also to see in BI if the heterogeneity improves.

Figure 5a and the results presented in the discussion need to be moved to the results section- you can then discuss in detail in the discussion but you can not introduce new data in the discussion.

I would have expected a fuller discussion of the potential limitations with respect to end effector robots and the reduced degrees of freedom at the ankle joint and its potential to limit balance and ankle strategy training  during RAGT as a possible explanation of the findings that warrant further investigation also.

As previously mentioned you need to address limitations in your own search and inclusion criteria at the end also.

Final proofing by a native english speaker would improve the manuscript at the end- while overall it is well written and the standard is high errors for example pg 2 line 63 9tough). line 96 (whit)pg 3 line 102 (Not); line 115 (remotion = removal); pg 4 line 171 (in Japan); pg 6 line 224 (besides- remove) ; pg 14 431 old stroke adults needs to be reworded

Author Response

Dear Reviewers, thanks for all your comments and contributes and for the possibility to improve the manuscript. Please find in red in this document the answer to your comments.

Reviewer 2

Dear authors 

It was with great interest that I read your systematic review of over-ground RAGT to affect change in balance after stroke . Indeed over-ground systems require initiation of movement and engage the vestibular system and additional postural control pathways in a way over-treadmill walking doesn't. There is good evidence of an altered neurological basis of treadmill walking at muscle (EMG) and brain (EEG)  systems so I was interested to read your review.

Your introduction sets the scene well and creates a good case for conducting your review.

You registered your review and have conducted it with good scientific rigour in keeping with the guidelines. The search is still current to Nov 2021.

The search terms as presented appear a little basic and do not include controlled vocabularly terms in databases. The single use of balance for your outcome is limited as it may miss for example equilibrium, centre of gravity/mass, postural control etc. and you did not search for "electromechanically assisted gait" in your robotic search string. You will need to identify these as study limitation at the end along with the decision to look for only english language studies. You may have missed additional studies because of these choices.
Thank you for your comment, we added this information as a limitation of the study. We reported this information as follow: “The search string and the inclusion of only English-language studies may have resulted in missing additional studies available in the literature.”

You have chosen to report 3 quality assessment tools and this is excessive I think. I would be inclined to report the D&B for all study types and the cochrane ROB for RCTs. 
Dear reviewer, according to your comment we have removed from the text the section on the PEDro assessment, including the Table 2. Consequently, we have renumbered the Tables and all their references in the whole text.

In your data synthesis section please clarify that it is a minimum of 3 RCTs with the same outcome measures for meta-analysis.
Thank you for allowing us to clarify. We have modified the sentence adding this information.

The next area of confusion is what you are labelling RCTs you say there are 3 and then include 6 in the Cochrane ROB assessment and use these in the meta-analysis. Looking at all 6 of these studies -they should all be classified as RCTs- this would make this section a lot cleaner and allow you -in the narrative synthesis -to discuss and report by level of evidence, RCTs first , then pilot studies (identifying them by methodology better- eg prospective cohort study/ non-randomised case control study etc as appropriate), then case report. In table one you label the Kotov studies as clinical prospective studies yet they have 2 groups that are randomised and you use a ROB tool for RCTs later.
Dear Reviewer, we are apologize for the misunderstanding. We initially decided to report the study design as reported and described by the Authors of each article. However, as you indicated, this heterogeneous nomenclature leads to confusion and for this reason, after a further careful control of the full-texts, we have modified the study designs in RCT as you suggested. Consequently, we have made changes in the narrative synthesis according to this improved organization.

I would delete table 2 and PEDro rating here and in figure 2 identify ROB tool for RCTs.
As already declared above, we removed from the text the section on the PEDro assessment. Also, the table 2 was deleted. Consequently, as already declared above, we have renumbered the Tables and their references in the whole text.

In 3.6 outcome measure- you need to provide detail with respect to what instrumentation measures were used for balance and again provide greater synthesis by study type rather than reporting each study seperately
Thank you for the possibility to improve this section. According to this and also previous comments we modified the organization of the reporting, starting from studies with higher level of evidence and providing a greater synthesis than reporting each study separately.

In figure 3(b) please remove the decimal place for the discrete variable of number of studies. Add a legend for COP; FRT etc so the figure can stand alone in its interpretation.
Thank you for these suggestions. We removed the decimal from the figure and added the missing legend.

In your meta-analysis figure 4 please identify on the label that it is a comparison of 0-RAGT with or without conventional therapy vs conventional therapy. I would advise you to do a further sensitivity analysis  by removing the Kotov study and looking at o-RAGT with CT vs CT alone, reporting the analysis results on the 2 pooled studies and also to see in BI if the heterogeneity improves.
Thank you for your comment. We added the information “with or without CT” in the label. Furthermore, we conducted further analysis for both BBS and BI outcomes by removing Kotov study as you suggested. However, no substantial differences were evidenced, and the results are very similar. Please find attached here the figures of these analysis:

BBS:

BI:

Figure 5a and the results presented in the discussion need to be moved to the results section- you can then discuss in detail in the discussion but you cannot introduce new data in the discussion.
Dear reviewer, thank you for the possibility to improve these sections. We decided to insert the Figure 5 in order to facilitate reading but after your comment, we understood that the data reported in figure 5 are the same of the table of demographic and clinical features data of participants. For this reason, we have decided to remove this figure as it was redundant. Some minor modifications were carried out in this section.

I would have expected a fuller discussion of the potential limitations with respect to end effector robots and the reduced degrees of freedom at the ankle joint and its potential to limit balance and ankle strategy training during RAGT as a possible explanation of the findings that warrant further investigation also.
Thank you for this comment, we agree with you. We have added a specific section on this topic in the discussion. Three new references were added (Li S. Ankle and Foot Spasticity Patterns in Chronic Stroke Survivors with Abnormal Gait. Toxins (Basel). 2020 Oct 7;12(10):646. doi: 10.3390/toxins12100646; Gorst T, Lyddon A, Marsden J, Paton J, Morrison SC, Cramp M, Freeman J. Foot and ankle impairments affect balance and mobility in stroke (FAiMiS): the views and experiences of people with stroke. Disabil Rehabil. 2016;38(6):589-96. doi: 10.3109/09638288.2015.1052888. Chen G, Patten C, Kothari DH, Zajac FE. Gait differences between individuals with post-stroke hemiparesis and non-disabled controls at matched speeds. Gait Posture. 2005 Aug;22(1):51-6. doi: 10.1016/j.gaitpost.2004.06.009). This section was reported as follow: “Furthermore, in the area of RAGT, to date there are no studies comparing the effects of treatments with end-effector versus o-RAGT devices. This lack of data in the literature leaves open a very interesting point of discussion. In fact, the main difference between end-effector and overground devices, is the origin of motion. In the case of end-effectors, the motion is generated by the device starting from the periphery while for the other devices it is the exoskeleton itself that generates the motion in all the joints. This difference is intriguing because most patients with stroke experience ankle-foot disability [55]. Following stroke, foot deformity, altered plantar sensory inputs, reduced ankle proprioception, altered motor control or toe clawing have all been observed, identifying a relationship between these impairments and balance impairments [56]. Specific foot and ankle impairments may also negatively contribute to perceptions of physical appearance and self-esteem as well as the quality of life was severely affected in stroke survivors, specifically in walking independently, due to the reduced peak of the ankle dorsiflexion angle in the paretic leg [57]. Future studies aimed to compare different type of s-RAGT vs o-RAGT may shed light on this aspect not yet investigated to the present time.”

As previously mentioned you need to address limitations in your own search and inclusion criteria at the end also.
As already reported above, we have added these limitations as you suggested.

Final proofing by a native english speaker would improve the manuscript at the end- while overall it is well written and the standard is high errors for example pg 2 line 63 9tough). line 96 (whit)pg 3 line 102 (Not); line 115 (remotion = removal); pg 4 line 171 (in Japan); pg 6 line 224 (besides- remove) ; pg 14 431 old stroke adults needs to be reworded
Thank you for pointing out some errors and for the possibility to correct the entire paper. We had the paper totally reviewed by a fellow English native speaker, rewriting some sections and correcting the errors pointed out in the comment.

Round 2

Reviewer 1 Report

Thank you for addressing the comments. 

My only comment is PRISMA flowchart. It stills without the included study in the meta-analysis.

Author Response

Thank you, we have modified the figure 1 accordingly.

Best